# Optimal Open-Loop Control of Discrete Deterministic Systems by Application of the Perch School Metaheuristic Optimization Algorithm

**Andrei V. Panteleev * and Anna A. Kolessa**

Department of Mathematics and Cybernetics, Moscow Aviation Institute (National Research University), 4, Volokolamskoe Shosse, 125993 Moscow, Russia; annacom84kolessa@yandex.ru
* Correspondence: avpanteleev@inbox.ru

**Abstract:** A new hybrid metaheuristic method for optimizing the objective function on a parallelepiped set of admissible solutions is proposed. It mimics the behavior of a school of river perch when looking for food. The algorithm uses the ideas of several methods: a frog-leaping method, migration algorithms, a cuckoo algorithm and a path-relinking procedure. As an application, a wide class of problems of finding the optimal control of deterministic discrete dynamical systems with a nonseparable performance criterion is chosen. For this class of optimization problems, it is difficult to apply the discrete maximum principle and its generalizations as a necessary optimality condition and the Bellman equation as a sufficient optimality condition. The desire to extend the class of problems to be solved to control problems of trajectory bundles and stochastic problems leads to the need to use not only classical adaptive random search procedures, but also new approaches combining the ideas of migration algorithms and swarm intelligence methods. The efficiency of this method is demonstrated and an analysis is performed by solving several optimal deterministic discrete control problems: two nonseparable problems (Luus–Tassone and LiHaimes) and five classic linear systems control problems with known exact solutions.

**Keywords:** bio-inspired algorithms; metaheuristic; optimal control; discrete dynamical system; migration algorithm; perch; population

## 1. Introduction

Global optimization methods are widely used in engineering calculations when designing complex technical systems, problems of information processing, decision making and optimal control, machine learning, etc. [1–7]. As a rule, the application of the necessary optimality conditions for finding the optimal open-loop control of discrete deterministic systems is associated with the solution of a boundary value problem for a system of difference equations, the computational complexity of which greatly increases with the state vector dimension. The use of quasi-gradient optimization methods often leads to obtaining locally optimal solutions. The use of direct optimization methods, such as the Luus–Jaacola method, makes it possible to obtain a result with the desired accuracy with an acceptable number of runs of the algorithm. Objective functions can be multi-extremal, nondifferentiable, difficult to calculate, or generally described by a black box model. Therefore, the application of necessary and sufficient optimality conditions, as well as quasi-gradient methods of the first and second orders, could be difficult. Among the zero-order methods, in which the search direction is determined only by information about the value of the objective function, two groups can be distinguished: stochastic metaheuristic algorithms [8–16] and deterministic methods of mathematical programming [17–19]. The former have gained popularity among applied mathematicians and engineers because they use various original heuristics driven by higher-level algorithms. Proof of convergence for these algorithms is usually complex and, thus, hard to provide, but they are able to obtain solutions of

sufficiently good quality in an acceptable time. The second group of methods is popular among mathematicians, since, under certain conditions, it can guarantee convergence to the global extremum. In the first group, it is conditionally possible to single out four well-known areas of research.

The first one includes evolutionary methods, in which the search process is associated with the evolution of a solutions population. These include Genetic Algorithms, Differential Evolution, Artificial Immune Systems, Covariance Matrix Adaptation Evolution Strategy, Scatter Search, Variable Mesh Optimization, Memetic Algorithms, Weed Colonization, Cuckoo Search, Self-Organizing Migrating Algorithms, etc.

The second area is represented by Swarm Intelligence algorithms: Particle Swarm Optimization, Artificial Bee Colony, Ant Colony Optimization, Bat-Inspired Algorithm, Cat Swarm Optimization, Firefly Algorithm, Grey Wolf Optimizer, Whale Optimization Algorithm, Bacterial Foraging Optimization, Fish School Search, Glowworm Swarm Optimization, Shuffled Frog-Leaping Algorithm, Krill Herd, Imperialist Competitive Algorithm, Stochastic Diffusion Search, Human Group Optimization Algorithm, etc. In this area, new original algorithms are being created and considered [20]. In the methods of this group, members of the populations exchange information during the search process and use information about the whole population leaders, local leaders among neighbors, and their own best results.

The third area is represented by physics-based algorithms: Central Force Optimization, Simulated Annealing, Adaptive Simulated Annealing, Harmony Search, Big Bang–Big Crunch, Spiral Dynamics Algorithm, Fireworks Algorithm, Grenade Explosion Method, Intelligent Water Drops Algorithm, Electromagnetism-like Mechanism, etc.

The fourth area is represented by multistart-based algorithms: Greedy Randomized Adaptive Search, Tabu Search, etc. In the four listed areas, a group of bio-inspired algorithms can be distinguished due to their nature-inspired interpretations [2,16].

This article is devoted to the development of the swarm intelligence group of methods based on observing the hunting of a school of river perches and hybridization mechanisms. Given the validity of the free lunch theorem [21], the problem of designing new optimization methods aimed at solving a certain class of applied problems remains relevant. The problem of creating new metaheuristic algorithms is important, since there is no universal optimization algorithm suitable for solving all problems of finding a global extremum. Therefore, almost constantly, new search procedures are proposed.

The aim of this work is to develop a new metaheuristic optimization algorithm that can be effective for solving a wide class of problems in optimal control theory. The metaheuristic algorithm proposed in this article can be applied to solve a wide class of problems of parametric optimization of continuous and discrete, deterministic and stochastic control systems in the presence of uncertainty in setting the initial conditions, parameters of the object model and the measuring system. In particular, the method should allow for solving both classical and nonclassical nonseparable optimization problems for nonlinear discrete deterministic dynamical control systems [1]. In addition to classical problems, problems of model predictive control theory could be solved [22].

A method for optimizing the objective function on a parallelepiped set of admissible solutions is proposed herein. It mimics the behavior of a school of river perch when looking for food. The method is hybrid because it uses the ideas of migration algorithms [23], methods that imitate the behavior of cuckoos [24] and frogs [25], and path-relinking procedures [26]. The advantage of the proposed optimization algorithm lies in the ability of feasible solutions to leave the area of attraction of local extremum points. This is achieved both through the main technique, which implements the movement of perches inside the perch cauldron, and through the high-level management of the use of well-established heuristic techniques such as Levy flights to explore distant areas of the set of feasible solutions, regular mixing of schools in order to exchange information, and path-relinking procedures.

The efficiency of this method is demonstrated and an analysis is performed by solving optimal deterministic discrete control problems: two nonseparable problems (Luus–Tassone and Li–Haimes) and five classic linear systems control problems with known exact solutions.

## 2. Materials and Methods

### 2.1. Open-Loop Control Problem

Let us consider a nonlinear dynamical system described by a difference equation

$$x(t+1) = f(t, \ x(t), \ u(t)), \tag{1}$$

where $t$ represents discrete time, $t \in T = [0, 1, \dots, N-1]$; the number of stages $N$ is given; $x$ is the state vector, $x \in \mathbf{R}^n$; $u$ is the control vector, $u \in U(t) \subseteq R^q$; $U(t)$ is the control values feasible set which, for each $t$, comprises the direct product of segments $[a_i(t), \ b_i(t)], i = 1, 2, \dots, q$; and $f(t, \ x, \ u) = (f_1(t, \ x, \ u), \dots, f_n(t, \ x, \ u))^T$ is a continuous vector function.

The initial state vector is given:

$$x(0) = x_0. \tag{2}$$

It is assumed that only the discrete time information $t$ is used in the control law, so open-loop control is considered.

The set of allowable processes $D(0, \ x_0)$ is a set of pairs $d = (x(\cdot), \ u(\cdot))$, where $x(\cdot) = \{x_0, \ x(1), \dots, x(N)\}$ is a trajectory, $u(\cdot) = \{u(0), \ u(1), \ \dots, \ u(N-1)\}$ is an admissible control, and $u(t) \in U(t)$, satisfying the state Equation (1) and initial condition (2).

The cost functional on the set $D(0, x_0)$ is defined as

$$I(d) = \sum_{t=0}^{N-1} f^0(t, \ x(t), \ u(t)) + F(x(N)), \tag{3}$$

or

$$I(d) = F(x(0), \dots, x(N); \ u(0), \dots, u(N-1)), \tag{4}$$

where $f^0(t, \ x, \ u)$, $F(x)$, $F(x(\cdot), \ u(\cdot))$ are given continuous functions.

It is required to find a pair $d^* = (x^*(\cdot), \ u^*(\cdot)) \in D(0, \ x_0)$ that minimizes the cost functional, i.e.,

$$I(d^*) = \min_{d \in D(0, x_0)} I(d). \tag{5}$$

The required trajectory $x^*(\cdot) = \{x_0, \ x^*(1), \dots, x^*(N)\}$ and control $u^*(\cdot) = \{u^*(0), \ u^*(1), \ \dots, \ u^*(N-1)\}$ (elements of a pair $d^*$) are the optimal trajectory and optimal control, correspondingly.

To solve the problem (5), an algorithm that imitates the behavior of a perch school is proposed. This algorithm belongs to the class of nature-inspired metaheuristic swarm intelligence algorithms [9,11,16]. This problem was also solved using the Sparrow Colony Optimization Method, with results close to those received in this paper [27].

### 2.2. Bio-Inspired Metaheuristic Optimization Method

The problem of finding the global extremum of the objective function $f(x) = f(x_1, \dots, x_n)$ on the set of feasible solutions of the form $D = [a_1, b_1] \times \cdots \times [a_n, b_n]$ is considered.

The Perch School Search method (PSS) imitates the behavior of a European river perch school. Medium-sized perches make schools of 5–12 individuals in order to find food. The smallest perches make schools of approximately 100 individuals. Generally, they take the fish-prey into a circle (perch boiler) and hold it there. Perches attack prey at the boiler edge, moving to its center. To find new food sources, perches use a migration mechanism. Usually, bigger perches swim at the depths, in holes and deep water, hunting by themselves. Rarely, they also make schools, like small ones do, in order to fight with other predatory fish (pikes and sanders). The river perch uses a very aggressive hunting model: it actively

pursues the prey, sometimes jumping out even on the surface of the water. The biggest perches are solitary and independent predators. The reason for this is that the big perches do not need to hunt collectively. These perches can hunt by themselves on any size fish available. In the case of feed places, lacking a regular habitat, perch begin to move and search for places full of small fish and other food.

Finite sets $I = \left\{ x^j = (x_1^j, x_2^j, \ldots, x_n^j)^T, j = 1, 2, \ldots, NP \right\} \subset D$ of possible solutions, named populations, are used for solving the problem of finding a global constrained minimum of an objective function, where $x^j$ is a perch-individual (potential feasible solution) with number $j$, and $NP$ is the population size.

In the beginning of the process, the method creates a population by the uniform distribution law on a feasible solution set, $D$, illustrated in Figure 1a. All solutions are ordered by increasing objective function value. After that, the created population is divided into several schools. The schools form a sequence: The best solution is placed in the first school, the next is placed in the second, etc. The $M$-th is placed in the $M$-th school, and the $(M + 1)$-th is then placed in the first school, and so on. This process is shown in Figure 1b. The described process corresponds to the information exchange among population members for an efficient approach to the global extremum.

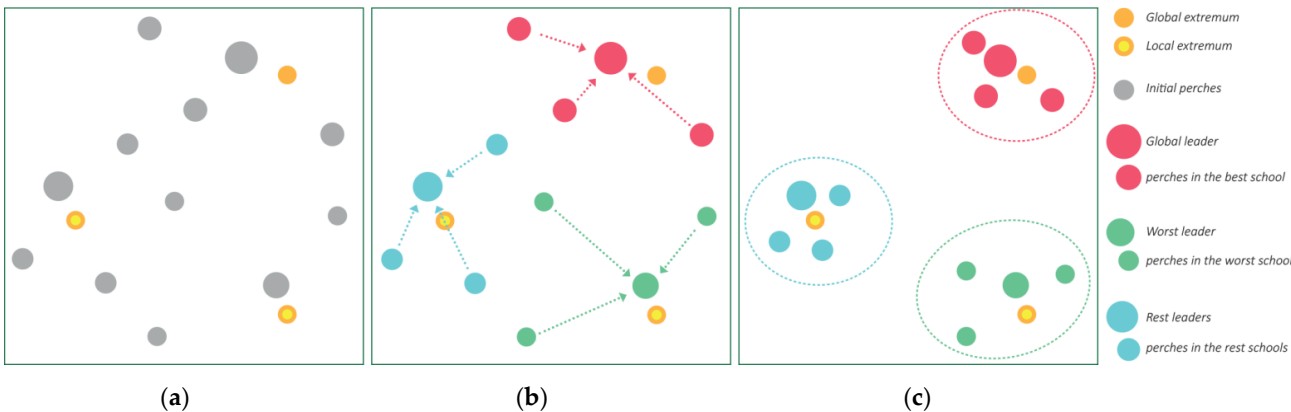

**Figure 1.** Hunting behavior of the river perch; (**a**) creation a population by the uniform distribution law on a feasible solution set, (**b**) separation of the whole population into schools, (**c**) realization of the perch boiler in each school.

In each school, the leader (the best possible solution in the school) is determined by the value of the objective function. Every school makes a perch boil in which there is a hunt, illustrated in Figure 1c. At that time, every perch in the school moves to its leader, exploring the boil edge. During this movement, perches remember their best position. As a result, new leaders' and members' positions for each school are found. Among the school leaders, the best and the worst leaders are found.

The perch boiler in the school with the best leader is realized, so perches of this school search the entire area in depth. As a result, the new absolute leader is found.

After that, among all schools, the school with the worst leader is chosen. This leader then moves into another area of the feasible solution set. For that, the school leader's movement is realized by a Lévy distribution with verification of belonging to the feasible solution set. The uniform distribution law on the parallelepiped set generates the remaining members of the school. A doubled distance from the leader to the nearest edge of set $D$ describes the size of the parallelepiped set. The received school realizes a perch boil and a new leader is found.

The rest of the schools swim in the direction of the current absolute leader of the whole feasible set. At this time, each local school leader moves in a straight line to the global leader. Other members of the school move parallel to the global leader. In this movement, all perches remember their best position and stay there.

In the end of the migration of all schools, the absolute leader is put into the *Pool* set. Then, the division of the entire population into schools begins again. Therefore, another new global iteration starts until it reaches a given number.

At the final step, leaders in the *Pool* set (identified at each global iteration of the algorithm) interact with each other. After a given number of times in the *Pool* set, three perches are chosen, and a path-relinking operation [26] is realized. As a result, this set is replenished by one more solution.

After the end of the path-relinking operation among the *Pool* set elements, the best answer is found, which is considered to be an approximate solution of the optimization problem.

The proposed method corresponds to hybrid algorithms because it contains ideas described in Shuffled Frog-Leaping Optimization (division into schools) [28], Cuckoo Search (Lévy flights) [29], Self-Organizing Migrating Algorithm (movement to the leader) [23] and the path-relinking algorithm [26] (searching in the *Pool* set).

Below is a detailed description of the algorithm.

**Step 1.** Set the parameters of the method:

- Controlling parameter $NStep$, which specifies number of steps before the end of the movement;
- Number of schools in the population, $M$;
- Number of perches in a school, $s$;
- Number of perches in the population, $NP = s \cdot M$;
- Stop parameter $Iter_{\max}$, which specifies the maximum number of iterations;
- Levy distribution parameter $\lambda$;
- Step size $\alpha$;
- Maximum number of path-relinking operations, $PR_{\max}$;
- Number of steps in the path-relinking operation, $\Delta_{pr}$.

**Step 2.** Creation of the initial perch population.

Step 2.1. Create population $I = \left\{ x^j = (x_1^j, x_2^j, \ldots, x_n^j)^T, j = 1, 2, \ldots, NP \right\} \subset D$ of $NP$ solutions (perches) with randomly generated coordinates $x_i$ from the segment $[a_i, b_i]$ using a uniform distribution:

$$x_i^j = a_i + rand_i[0,\ 1] \cdot (b_i - a_i), i = 1, \ldots, n;\ j = 1, \ldots, NP$$

where $rand_i[0,\ 1]$ is the uniform distribution law on the segment $[0; 1]$.

Step 2.2. For each solution (perch) in the population, calculate the value of the objective function.

Set the value: $iter = 0$ (global iteration count).

**Step 3.** Separation of the whole population into schools.

Step 3.1. Arrange the solutions in the population in ascending order of objective function values.

Step 3.2. Make $M$-many schools of $s$ perches each: The best solution (with the smallest value of the objective function) is put into the first school, the next is put into the second, etc. The $M$-th is put into the $M$-th school, and the $M + 1$-th is put into the first school, etc. As the result, there are $M$-many schools of $s$ perches each, so $NP = M \cdot s$. The first perch placed in a school is its leader $x^{loc,m}$, $m = 1, \ldots, M$. The leader of the first school is also the leader of the whole population (the global leader): $x^{loc,1} = x^{glob}$.

**Step 4.** Realization of the perch boiler in each school.

Step 4.1. For each school $m = 1, \ldots, M$, do the following.

Move each perch to the school's leader near its border:

$$x^{j,m,k} = x^{j,m} + k \frac{\left( x^{loc,m} - x^{j,m} \right)}{Nstep}\ ,\ k = 0, 1, \ldots, [\sigma \cdot Nstep];\ j = 1, \ldots, s;$$

where $x^{j,m}$ is the initial position of perch $j$ in school $m$; $x^{j,m,k}$ is a position of this perch during movement; $x^{loc,m}$ is a position of the school leader numbered $m$; $[\cdot]$ is the integer part of a number; and boiler parameter $\sigma \in [0,1;0,5]$ is generated via the uniform distribution law at each iteration for each school independently.

After all steps, find the best step for each perch (the step at which the value of the objective function was the smallest); the perch must be in this best position $x^{j,m,new}$:

$$x^{j,m,new} = \underset{k=0,1,\ldots,[\sigma \cdot Nstep]}{\arg\min} f\left(x^{j,m,k}\right), \ j = 1,\ldots,s.$$

Step 4.2. Choose a new leader in each school: $x^{loc,m,new}$, $m = 1,\ldots,M$.

Step 4.3. Arrange the schools in the population in ascending order of objective function values of its leaders. The global leader is in the first school: $x^{loc,1} = x^{glob}$; local leaders are in other schools: $x^{loc,m}$, $m = 2,\ldots,M$; and the maximum value of the objective function among leaders is in the school numbered $M$.

**Step 5.** School swimming with the global leader.

Step 5.1. Move each perch toward the global leader, moving along the line connecting them (approaching and then receding in the same direction):

$$x^{j,1,k} = x^{j,1} + k\frac{\left(x^{glob} - x^{j,1}\right)}{Nstep}, \ k = 0,1,\ldots,[\sigma_1 \cdot Nstep]; \ j = 1,\ldots,s;$$

where boiler parameter $\sigma_1 \in [1;1,5]$ is generated via the uniform distribution law at each iteration.

Step 5.2. After all steps, find the best step for each perch (the step at which the value of the objective function was the smallest); the perch must be in this best position $x^{j,1,new}$:

$$x^{j,1,new} = \underset{k=0,1,\ldots,[\sigma_1 \cdot Nstep]}{\arg\min} f\left(x^{j,1,k}\right), \ j = 1,\ldots,s.$$

Step 5.3. Choose a new leader in the school, $x^{loc,1,new} = x^{glob,new}$.

**Step 6.** School swimming with the worst leader.

Step 6.1. Leader swimming: A new position for the leader is randomly generated via the Levy distribution:

$$x_i^{loc,M,new} = x_i^{loc,M} + \frac{\alpha}{iter} \cdot Levy_i(\lambda), \ i = 1,\ldots,n,$$

where $x_i^{loc,M}$ is the position coordinates of the school's leader in the current iteration, $\alpha$ is the step size, and $\lambda \in (1;3]$. To generate a random variable according to the Levy distribution, the following is required:

For each coordinate $x_i = Levy_i(\lambda)$, generate a number $R_i$, $i = 1,\ldots,n$ via the uniform distribution law on the set $[\varepsilon; \ b_i - a_i]$, where $\varepsilon = 10^{-7}$ is the distinguishability constant, and do the following:

- Generate numbers $\theta_i = R_i \cdot 2\pi$ and $L_i = (R_i + \varepsilon)^{-\frac{1}{\lambda}}$, $i = 1,\ldots,n$, where $\lambda$ is a distribution parameter;
- Calculate values of coordinates using the formulas below:

$$x_i(\lambda) = L_i \sin \theta_i, \ i = 1,\ldots,\left[\frac{n}{2}\right]; \ x_i(\lambda) = L_i \cos \theta_i, \ i = \left[\frac{n}{2}\right] + 1,\ldots,n$$

If the obtained value of the coordinate $x_i$ does not belong to the set of feasible solutions, that is, $x_i \notin [a_i; b_i]$, then repeat the generation process for the coordinate $x_i$.

Step 6.2. Generate new positions of school members $x^{j,M}$ via the uniform distribution on a parallelepiped formed by the direct product of segments $[x_i^{loc,M} - \hat{x}_i, x_i^{loc,M} + \hat{x}_i]$, where $\hat{x}_i = \min\left\{ (x_i^{loc,M} - a_i), (b_i - x_i^{loc,M}) \right\}$.

Step 6.3. Make a perch boiler in the obtained school.

Move each perch to a school's leader near its border:

$$x^{j,M,k} = x^{j,M} + k\frac{\left(x^{loc,M,new} - x^{j,M}\right)}{Nstep}, k = 0, 1, \ldots, [\sigma_3 \cdot Nstep]; j = 1, \ldots, s;$$

where boiler parameter $\sigma_3 \in [0, 1; 0, 5]$ is generated via the uniform distribution law at each iteration.

After all steps, find the best step for each perch (the step at which the value of the objective function was the smallest); the perch must be in this best position $x^{j,M,new}$:

$$x^{j,M,new} = \underset{k=0,1,\ldots,[\sigma_3 \cdot Nstep]}{\arg\min} f\left(x^{j,M,k}\right), j = 1, \ldots, s.$$

Step 6.4. Choose a new leader for the school, $x^{loc,M}$.

**Step 7.** Swimming of the remaining schools.

For all schools where $m \in \{2, \ldots, M - 1\}$, do the following.

Step 7.1. Move the school's leader to the current absolute (best) leader:

$$x^{loc,m,k} = x^{loc,m} + k\frac{\left(x^{glob,new} - x^{loc,m}\right)}{Nstep}, k = 0, 1, \ldots, [\sigma_2 \cdot Nstep];$$

where boiler parameter $\sigma_2 \in [0, 6; 0, 8]$ is generated via the uniform distribution law at each iteration for each school, independently.

Step 7.2. Make the movement of other school members parallel to that of the absolute leader:

$$x^{j,m,k} = x^{j,m} + k\frac{\left(x^{glob,new} - x^{loc,m}\right)}{Nstep}, k = 0, 1, \ldots, [\sigma_2 \cdot Nstep]; j = 1, \ldots, s.$$

After all steps, find the best step for each perch (the step at which the value of the objective function was the smallest); the perch must be in this best position $x^{j,m,new}$:

$$x^{j,m,new} = \underset{k=0,1,\ldots,[\sigma_2 \cdot Nstep]}{\arg\min} f\left(x^{j,m,k}\right), j = 1, \ldots, s.$$

Step 7.3. Every school member must take their best position reached in the swimming process. Choose a new school leader with position $x^{loc,m}, m \in \{2, \ldots, M - 1\}$.

**Step 8.** Finding the new global leader of the population among the schools' leaders.

Choose the best solution among the solutions, corresponding to the leaders, and put it into the set *Pool*.

**Step 9.** Checking the stop conditions.

If $iter = Iter_{\max}$, go to Step 10. Otherwise, set $iter = iter + 1$ and go to Step 3.

**Step 10.** Intensive searching in the set *Pool*.

Step 10.1. Set the value $pr = 1$.

Step 10.2. Choose three different random solutions $x_{pool}^p, x_{pool}^q, x_{pool}^r$ in *Pool*.

Step 10.3. Find the solution

$$x_{pool}^{pq} = arg \underset{j=1,\ldots,\Delta_{pr}-1}{\min} f\left(x_{pool}^p + j\,(x_{pool}^q - x_{pool}^p)/\Delta_{pr}\right)$$

Step 10.4. Add the solution

$$x^{new} = arg \min_{j=1,...,\Delta_{pr}-1} f\left(x_{pool}^{pq} + j\,(x_{pool}^{r} - x_{pool}^{pq})/\Delta_{pr}\right)$$

into *Pool*. Set the value $pr = pr + 1$.

Step 10.5. If $pr > PR_{max}$, go to Step 11. Otherwise, go to Step 10.2.

**Step 11.** Choosing the best solution. Among the solutions in *Pool*, find the best one, which is considered to be an approximate solution of the optimization problem.

During the creation and testing of this algorithm, the following recommendations were developed for choosing parameters.

Parameter selections within the following ranges provide better results: controlling parameter $NStep \in [50;100]$; number of schools in the population $M \in [3;10]$; number of perches in a school $s \in [15;100]$; stop parameter $Iter_{max} \in [50;4000]$; maximum number of path-relinking operations $PR_{max} \in [10;20]$; number of steps in path-relinking operation $\Delta_{pr} \in [5;10]$; Levy distribution parameter $\lambda \in (1;3]$; step size $\alpha \in [0.2;0.6]$.

The proposed algorithm can be applied in the theory of optimal control using spectral and pseudospectral methods for representing trajectories and control laws in the form of expansions in basic systems with indeterminate coefficients.

## 3. Results

### 3.1. Example 1. The One-Dimensional Optimal Control Problem with an Exact Solution

The behavior of the control object is described by the difference equation:

$$x(t+1) = x(t) + u(t),$$

where $x \in \mathbf{R}$; $t = 0, 1, \ldots, N-1$; $u \in \mathbf{R}$.

The cost functional on the feasible set of processes is:

$$I = \tfrac{1}{2} \sum_{t=0}^{N-1} \gamma^{-t} u^2(t) + x(N), \qquad \gamma > 1 \ .$$

The problem is to find the minimum value of the cost functional and optimal process $(x^*(\cdot),\ u^*(\cdot))$.

Analytical solution:   $u^*(t) = -\gamma^t$;   $x^*(t) = x_0 + \dfrac{(1-\gamma^k)}{\gamma-1}$;   $\min I = x_0 + \dfrac{1-\gamma^N}{2(\gamma-1)}$.

Initial condition: $x(0) = 0$; control constraint: $-2 \cdot 10^4 \le u \le 2 \cdot 10^4$. Number of stages: $N = 50$. The exact value of the functional is $\min I = -581.954264398477$.

The obtained approximate value of the functional was $I^* = -581.954264313551$, with a runtime of 5.36 min. We used C# (7.2) with NET Framework (4.7.2), Windows Forms to implement this algorithm and solve the optimal control problems. A 3.3 GHz Intel Core i7-2860QM was used to solve this problem and the others below. Table 1 shows the parameters of the method for solving this problem. Figure 2 shows the obtained control and corresponding trajectory.

**Table 1.** Parameters of PSS for solving the problem in Example 1.

| *NStep* | *M* | *s* | *Iter*$_{max}$ | $\lambda$ | $\alpha$ | *PR*$_{max}$ | $\Delta_{pr}$ |
|---|---|---|---|---|---|---|---|
| 50 | 6 | 50 | 4000 | 1.4 | 0.2 | 20 | 9 |

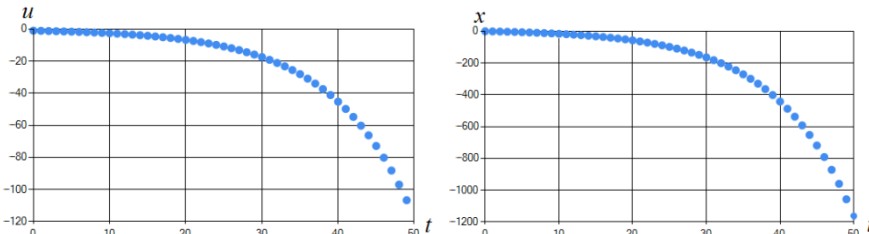

**Figure 2.** Obtained optimal control and corresponding optimal trajectory in Example 1.

To establish the relationship between the parameters and the effectiveness of the method, we solved the problem with other parameters of the method, which are shown in Table 2. The initial condition, control constraint, and number of stages were the same. Figure 3 shows the obtained control and corresponding trajectory.

**Table 2.** Parameters of PSS for solving the problem in Example 1.

| *NStep* | *M* | *s* | *Iter*$_{max}$ | $\lambda$ | $\alpha$ | *PR*$_{max}$ | $\Delta_{pr}$ |
|---|---|---|---|---|---|---|---|
| 50 | 6 | 25 | 1000 | 1.4 | 0.2 | 20 | 9 |

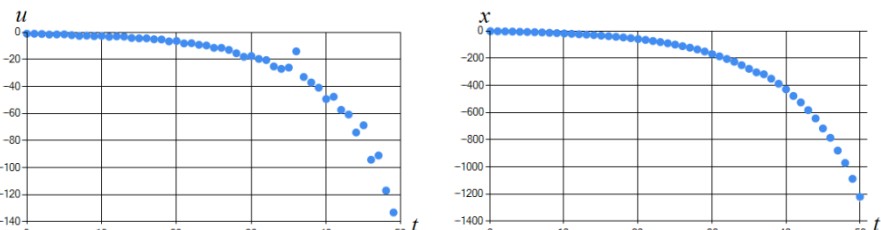

**Figure 3.** Obtained optimal control and corresponding optimal trajectory in Example 1.

The obtained approximate value of the functional was $I^* = -569.064908725205$, and the runtime was 0.31 min. In this case, the obtained approximate value of the functional was worse than that before because of the different parameters.

Below is a solution of the same problem from Example 1, but the number of stages was $N = 80$; this is shown in order to demonstrate the growth of time for finding the solution. The parameters of the method are shown in Table 1. Figure 4 shows the obtained control and corresponding trajectory.

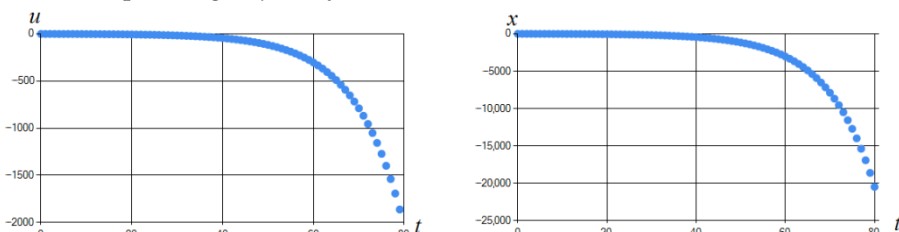

**Figure 4.** Obtained optimal control and corresponding optimal trajectory in Example 1.

The exact value of the functional is $\min I = -10237.001072927329$. The obtained approximate value of the functional was $I^* = -10237.0010729268$, and the runtime was 09:31 min.

Example 1 demonstrates the possibility of solving optimal control problems for economic models with a quality criterion that includes discounting. A similar problem occurs in machine learning problems such as deep reinforcement learning.

### 3.2. Example 2. Luus–Tassone Nonseparable Control Problem

The behavior of the control object is described by the system of difference equations:

$$x_1(t+1) = \frac{x_1(t)}{1+0{,}01u_1(t)(3+u_2(t))};$$
$$x_2(t+1) = \frac{x_2(t)+u_1(t)x_1(t+1)}{1+u_1(t)(1+u_2(t))};$$
$$x_3(t+1) = \frac{x_3(t)}{1+0{,}01u_2(t)(1+u_3(t))},$$

where $x \in \mathbf{R}^3$, $t = 0, \ldots, N-1$, and $u \in \mathbf{R}^3$.

The cost functional on the feasible set of processes is:

$$I = {x_1}^2(N) + x_2^2(N) + x_3^2(N)+$$
$$+\left[\left(\sum_{k=1}^{N} x_1^2(k-1) + x_2^2(k-1) + 2u_3^2(k-1)\right)\left(\sum_{k=1}^{N} x_3^2(k-1) + 2u_1^2(k-1) + 2u_2^2(k-1)\right)\right]^{\frac{1}{2}}.$$

The problem is to find the minimum value of the cost functional and optimal process $(x^*(\cdot),\ u^*(\cdot))$.

Initial state vector $x(0) = (2;\ 5;\ 7)^T$; control constraints: $0 \le u_1(t) \le 4$; $0 \le u_2(t) \le 4$; $0 \le u_3(t) \le 0.5$. Number of stages: $N = 20$.

The best known value of the cost functional obtained for this problem is $\min I = 209.26937$ [1].

The obtained approximate value of the functional was $I^* = 209.429533683522$, and the runtime was 49.92 s. Table 3 shows the parameters of the method for solving this problem. Figures 5 and 6 show the obtained control and corresponding trajectory.

**Table 3.** Parameters of PSS for solving the problem in Example 2.

| *NStep* | *M* | *s* | *Iter*$_{max}$ | $\lambda$ | $\alpha$ | *PR*$_{max}$ | $\Delta_{pr}$ |
|---------|-----|-----|----------------|-----------|----------|--------------|---------------|
| 100 | 3 | 50 | 900 | 1.4 | 0.3 | 15 | 7 |

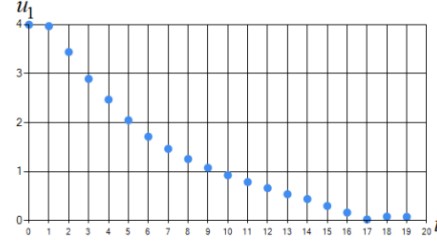 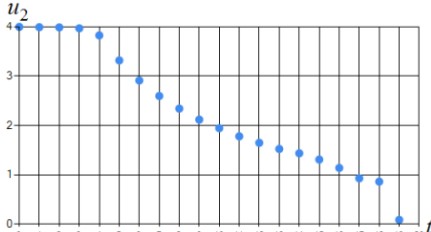 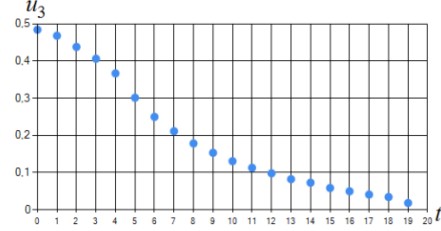

**Figure 5.** Obtained optimal control in Example 2.

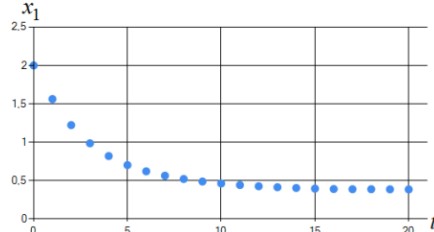  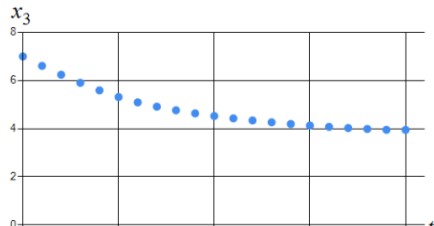

**Figure 6.** Obtained optimal trajectory in Example 2.

### 3.3. Example 3. Li–Haimes Nonseparable Control Problem

The behavior of the control object is described by difference equations:

$$x(1) = x(0)^{u(0)}; \quad x(2) = (1+u(1)) \cdot x(1); \quad x(3) = x(2) + u(2),$$

where $x \in \mathbf{R}$, $t = 0, 1, 2$, and $u \in \mathbf{R}$.

The cost functional on the feasible set of processes is:

$$I = [x^2(0) + x^2(1) + (2x^2(2) + x^2(3)) \exp(x^2(1))] \times [50 + u^2(0) + (u^2(1) + u^2(2)) \exp(u^2(0))]^{1/2}.$$

The problem is to find the minimum value of the cost functional and optimal process $(x^*(\cdot), \; u^*(\cdot))$.

Initial condition: $x(0) = 15$; control constraint: $-1 \le u(t) \le 1$, $t = 0, 1, 2$. Number of stages: $N = 3$. The best known value of the cost functional obtained for this nonseparable control problem is $\min I = 1596.4796778$ [1].

The obtained approximate value of the functional was $I^* = 1596.47967783389$, with a runtime of 00.19 s. Table 4 shows the parameters of the method for solving this problem. Figure 7 shows the obtained control and corresponding trajectory; the numerical solution of the problem is presented in Table 5.

**Table 4.** Parameters of PSS for solving the problem in Example 3.

| *NStep* | *M* | *s* | *Iter*$_{max}$ | $\lambda$ | $\alpha$ | *PR*$_{max}$ | $\Delta_{pr}$ |
|---------|-----|-----|----------------|-----------|----------|--------------|---------------|
| 100 | 4 | 20 | 65 | 3 | 0.6 | 15 | 5 |

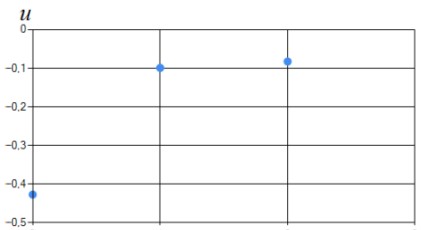 

**Figure 7.** Obtained optimal control and corresponding optimal trajectory in Example 3.

**Table 5.** Obtained optimal control and optimal trajectory (numerically).

| *t* | *u* | *x* |
|-----|-----|-----|
| 0 | −0.42716 | 15.00000 |
| 1 | −0.09897 | 0.31450 |
| 2 | −0.08238 | 0.28337 |
| 3 | | 0.20099 |

Examples 2 and 3 are devoted to solving a class of optimal control problems with nonseparable cost functionals, for which it is difficult to apply necessary and sufficient optimality conditions for discrete deterministic dynamical systems.

### 3.4. Example 4. The Two-Dimensional Lagrange Optimal Control Problem with an Exact Solution

The behavior of the control object is described by a system of difference equations:

$$x_1(t+1) = x_1(t) + u(t); \quad x_2(t+1) = 2x_1(t) + x_2(t),$$

where $x \in \mathbf{R}^2$, $t = 0, 1$, and $u \in \mathbf{R}$.

The cost functional on the feasible set of processes is:

$$I = \sum_{t=0}^{1} \left[ x_1{}^2(t) + x_2^2(t) + u^2(t) \right]$$

The problem is to find the minimum value of the cost functional and optimal process $(x^*(\cdot), \; u^*(\cdot))$.

Initial state vector $x(0) = (2;1)^T$; control constraint: $-10^5 \le u \le 10^5$. Number of stages: $N = 2$. The exact solution could be found as $u^* = \{-1;0\}$; $x_1^*(\cdot) = \{2;1;1\}$; $x_2^*(\cdot) = \{1;5;7\}$. The exact value of the cost functional is $\min I = 32$.

The obtained approximate value of the functional was $I^* = 32$, and the runtime was 01.58 s. Table 6 shows the parameters of the method for solving this problem. Figure 8 shows the obtained control and corresponding trajectory.

**Table 6.** Parameters of PSS for solving the problem in Example 4.

| NStep | M | s | Iter$_{max}$ | $\lambda$ | $\alpha$ | PR$_{max}$ | $\Delta_{pr}$ |
|---|---|---|---|---|---|---|---|
| 100 | 4 | 100 | 110 | 3 | 0.6 | 15 | 5 |

 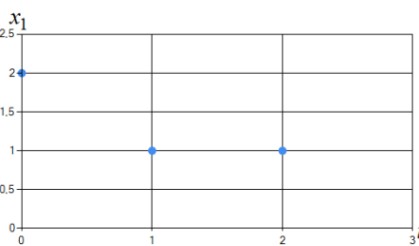 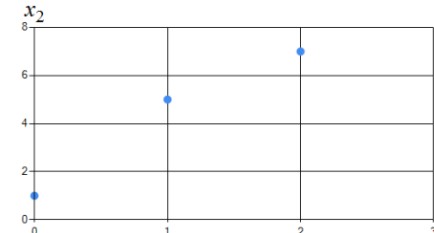

**Figure 8.** Obtained optimal control and corresponding optimal trajectory in Example 4.

### 3.5. Example 5. The Two-Dimensional Meyer Optimal Control Problem with an Exact Solution

The behavior of the control object is described by a system of difference equations:

$$x_1(t+1) = x_2(t); \quad x_2(t+1) = x_1(t) - u(t),$$

where $x \in \mathbf{R}^2$, $t = 0, 1$, and $u \in \mathbf{R}$.

The cost functional on the feasible set of processes is:

$$I = x_1{}^2(2) + x_2^2(2)$$

The problem is to find the minimum value of the cost functional and optimal process $(x^*(\cdot), \ u^*(\cdot))$.

Initial state vector $x(0) = (2;-3)^T$; control constraint: $-1 \le u \le 1$. Number of stages: $N = 2$. The exact value of the cost functional is $\min I = 5$.

The obtained approximate value of the functional was $I^* = 5.00000000000002$, with a runtime of 00.25 s. Table 7 shows the parameters of the method for solving this problem. Figure 9 shows the obtained control and corresponding trajectory; the numerical solution of the problem is presented in Table 8.

**Table 7.** Parameters of PSS for solving the problem in Example 5.

| NStep | M | s | Iter$_{max}$ | $\lambda$ | $\alpha$ | PR$_{max}$ | $\Delta_{pr}$ |
|---|---|---|---|---|---|---|---|
| 100 | 4 | 15 | 120 | 1.5 | 0.4 | 10 | 5 |

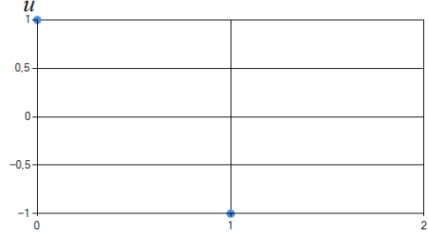  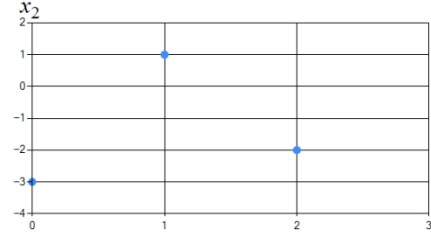

**Figure 9.** Obtained optimal control and corresponding optimal trajectory in Example 5.

**Table 8.** Obtained optimal control and optimal trajectory (numerically).

| $t$ | $u$ | $x_1$ | $x_2$ |
|-----|-----|-------|-------|
| 0 | 1 | 2 | −3 |
| 1 | −1 | −3 | 1 |
| 2 | | 1 | −2 |

Examples 4 and 5 demonstrate the application of the proposed method to a class of linear-quadratic control problems.

### 3.6. Example 6. The Two-Dimensional Bolza Optimal Control Problem with an Exact Solution

The behavior of the control object is described by a system of difference equations:

$$x_1(t+1) = x_2(t); \quad x_2(t+1) = 2x_2(t) - x_1(t) + \frac{1}{N^2}u(t),$$

where $x \in \mathbf{R}^2$, $t = 0, 1, \ldots, N-1$, and $u \in \mathbf{R}$.

The cost functional on the feasible set of processes is:

$$I = -x_1(N) + \frac{1}{2N}\sum_{t=0}^{N-1} u^2(t)$$

The problem is to find the minimum value of the cost functional and optimal process $(x^*(\cdot), u^*(\cdot))$.

Analytical solution: $u^*(t) = \frac{N-t-1}{N}$; $\min I = -\frac{1}{3} + \frac{3N-1}{6N^2} + \frac{1}{2N^3}\sum_{t=0}^{N-1} t^2$.

Initial state vector $x(0) = (0; 0)^T$; control constraint: $0 \leq u \leq 100$. Number of stages: $N = 10$. The exact value of the cost functional is $\min I = -0.1425$.

The obtained approximate value of the functional was $I^* = -0.142499999999796$, with a runtime of 07.39 s. Table 9 shows the parameters of the method for solving this problem. Figures 10 and 11 show the obtained control and corresponding trajectory.

**Table 9.** Parameters of PSS for solving the problem in Example 6.

| *NStep* | *M* | *s* | *Iter*$_{max}$ | $\lambda$ | $\alpha$ | *PR*$_{max}$ | $\Delta_{pr}$ |
|---------|-----|-----|----------------|-----------|----------|--------------|---------------|
| 100 | 3 | 30 | 750 | 1.7 | 0.6 | 10 | 5 |

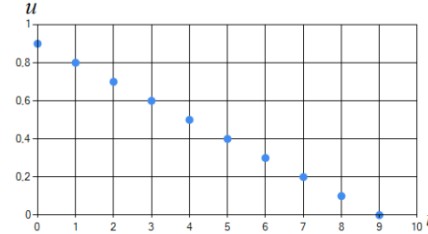 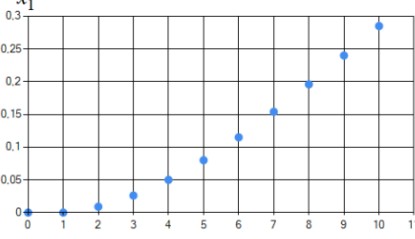 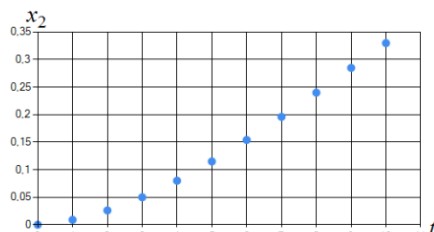

**Figure 10.** Obtained optimal control and corresponding optimal trajectory in Example 6.

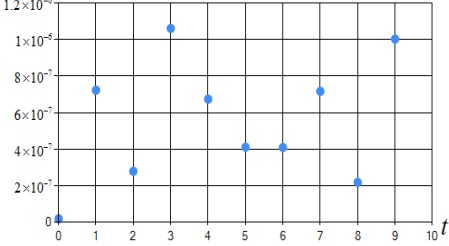

**Figure 11.** Control discrepancy at each moment of discrete time.

### 3.7. Example 7. The Two-Dimensional Meyer Optimal Control Problem with an Exact Solution

The behavior of the control object is described by a system of difference equations:

$$x_1(t+1) = x_1(t) + 2u(t); \quad x_2(t+1) = -x_1^2(t) + x_1(t) + u^2(t),$$

where $x \in \mathbf{R}^2$, $t = 0, 1$, and $u \in \mathbf{R}$.

The cost functional on the feasible set of processes is: $I = -x_2(2)$.

The problem is to find the minimum value of the cost functional and optimal process $(x^*(\cdot), u^*(\cdot))$.

Initial state vector $x(0) = (3; 0)^T$; control constraint: $-5 \leq u \leq 5$. Number of stages: $N = 2$. The exact value of the cost functional is $\min I = -19$. Note that the discrete maximum principle does not hold for this example.

This problem has two solutions: $x_1^{1*}(\cdot) = \{3; -1; 9\}$, $x_2^{1*}(\cdot) = \{0; -5; 19\}$, $u^{1*}(\cdot) = \{-2; 5\}$ and $x_1^{2*}(\cdot) = \{3; -1; -11\}$, $x_2^{2*}(\cdot) = \{0; -5; 19\}$, $u^{2*}(\cdot) = \{-2; -5\}$; the algorithm finds each of them.

The obtained approximate values of the functional were $I^{1*} = -18.9999999999354$, with a runtime of 00.30, and $I^{2*} = -18.9999999999939$, with a runtime of 00.31 s. Table 10 shows the parameters of the method for solving this problem. Figures 12 and 13 show the obtained control and corresponding trajectory (for the first and second solutions, correspondingly); the numerical solutions of the problem are presented in Tables 11 and 12.

**Table 10.** Parameters of PSS for solving the problem in Example 7.

| NStep | M | s | Iter$_{max}$ | $\lambda$ | $\alpha$ | PR$_{max}$ | $\Delta_{pr}$ |
|---|---|---|---|---|---|---|---|
| 100 | 6 | 20 | 75 | 1.3 | 0.2 | 10 | 5 |

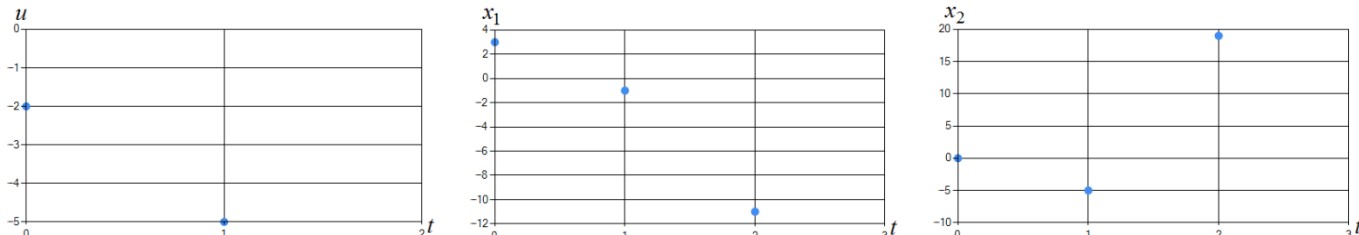

**Figure 12.** Obtained optimal control and corresponding optimal trajectories for the first solution to Example 7.

**Figure 13.** Obtained optimal control and corresponding optimal trajectories for the second solution to Example 7.

**Table 11.** Obtained optimal control and optimal trajectories for the first solution (numerically).

| $t$ | $u$ | $x_1$ | $x_2$ |
|---|---|---|---|
| 0 | $-2$ | 3 | 0 |
| 1 | 5 | $-0.99999$ | $-5.00002$ |
| 2 | | 9.00001 | 19 |

**Table 12.** Obtained optimal control and optimal trajectories for the second solution (numerically).

| $t$ | $u$ | $x_1$ | $x_2$ |
|---|---|---|---|
| 0 | −2 | 3 | 0 |
| 1 | −5 | −1 | −5 |
| 2 | | −11 | 19 |

## 4. Conclusions

In this paper, we proposed a new bio-inspired metaheuristic optimization algorithm, Perch School Search, for finding the solution to open-loop control problems for discrete deterministic dynamical systems.

The proposed algorithm combines the ideas of several well-known metaheuristic algorithms, providing an appropriate-quality solution for many practical optimization tasks.

The efficiency was shown in Examples 1–7, where the values of the cost functional obtained via the algorithm were equal or close to the analytical solution or to the best known results.

The direction of further development of this work is the application of the proposed global optimization metaheuristic algorithm to the problems of optimal control of bundles of trajectories of nonlinear deterministic discrete dynamical systems, discrete stochastic systems, and systems of joint estimation and control.

**Author Contributions:** Conceptualization, A.V.P.; methodology, A.V.P. and A.A.K.; software, A.A.K. All authors have read and agreed to the published version of the manuscript.

**Funding:** This research received no external funding.

**Institutional Review Board Statement:** Not applicable.

**Informed Consent Statement:** Not applicable.

**Data Availability Statement:** Source code at https://github.com/AeraConscientia/N-dimensional PerchOptimizer (accessed on 21 March 2022).

**Conflicts of Interest:** The authors declare no conflict of interest.

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
