# Peer review of "Optimal Open-Loop Control of Discrete Deterministic Systems by Application of the Perch School Metaheuristic Optimization Algorithm"

_algorithms, doi:10.3390/a15050157_

Round 1
Reviewer 1 Report
The issues discussed in this paper need further improvements in the following several aspects:
- The theoretical contributions should be stressed in detail in Introduction.
- Advantages of the proposed algorithm upon the well-known algorithms should be stressed.
- In introduction, it is not enough to state the current work. It should be expended and reconstructed. Including the motivation, the main difficulties, the main work and the improvements compared with previous related works should be emphasized in this section.
- I suggest comparing the simulations with the results of the recent (2021-2022) related valid references.
- The methodologies that can also guarantee the optimal control, that is, Adaptive Optimal Multi-Surface Back-Stepping Sliding Mode Control Design for the Takagi-Sugeno Fuzzy Model of Uncertain Nonlinear System With External Disturbance; A Lyapunov-Based Optimal Integral Finite-Time Tracking Control Approach for Asymmetric Nonholonomic Robotic Systems; A New Event-Triggered Type-3 Fuzzy Control System for Multi-Agent Systems: Optimal Economic Efficient Approach for Actuator Activating, should be discussed in the Introduction part.
- The importance of the problem considered in this paper should be further addressed.
- The directions to further and improve the work should be added as future recommendation section after ‘conclusions’ section.
- The types of software employed for solving the problem and also simulation experiments are unclear.
- Please check carefully all notations and equations.
Reviewer 2 Report
GENERAL PURPOSE: The paper proposes a new hybrid metaheuristic optimization method for solving optimization problems. It is built gathering techniques like frog-leaping, migration algorithms with the main idea: the river perch aim of looking for food. The methodology is promissory and it can call attention on Control Theory Community. According with the paper, the proposal works well for convex sets of solutions.
COMMENTS
Abstract: Authors must include the scientific gap filled with this algorithm: i.e. it is need to offer more metaheuristic methods for solving current problems in which traditional methods fail.
INTRODUCTION:
In line 32. Authors must be carefully with comments like “the convergence of these methods cannot be proven”. It is preferred to say: proof of convergence for this algorithms are usually complex and so hard to make.
Line 60-62. Good description for the focus of the optimization proposal.
Line 64. Good description of the GAP. You should expand this description in order to offer call the attention of researchers in Control Theory Area.
MATERIALS AND METHODS:
Authors must use adequate nomenclature for discrete problems in control theory.
In line 77. It is better if author say: T=[ , 1 … , N-1] represents the sample number.
Authors should include come images in order to illustrate the main ideas behind the behavior of river perch hunting. It could illustrate better the method.
Author could include some phrases in order to finalize properly the section of materials and methods. It could help some discussion around the parameters that could be tuned in order to reach good solutions.
Also, authors can discuss the features of the optimization problems in the case of control theory.
RESULTS:
Authors are invited to discuss or to offer some comments about the Example 1, Example 2. This is because they could represent some family of problems that can be solved by this methodology.
CONCLUSIONS:
Authors can offer additional explanations on the third part of the conclusion in order to offer more contributions from the methodology for solving optimal control problems. Those explanations can be about computational burden, exponential growing of time with higher number of time steps, solutions so close to analytical solutions.
REFERENCES:
References are adequate, actualized and relevant.
In general the article is well written. Reader can understand the text. I suggest some images in order to offer more clarity about the paper. May be, authors can sacrifice some examples and explain or discuss a bit more the features of them in order to gain more knowledgement from those early works. Authors should include variations on parameters for one single example in order to stablish relations between parameters and the performance of the method.
Excellent proposal using the river perch strategy. Congrats.
Round 2
Reviewer 1 Report
the paper is revised well.